# A food-grade cell dissociation agent via regulatory pre-check framework

**Hiroaki Hatano, Misaki Sawada, Misaki Kitsukawa, Azusa Hayakawa, Hiroaki Kondo, Ikko Kawashima** [ORCID] *

Production Development Department, IntegriCulture Inc., Fuzisawa City, Kanagawa Prefecture, Japan

* ikko@integriculture.com

## Abstract

Advancements in cell-based foods require materials that meet the highest food safety standards. Conventional cell dissociation agents, such as animal-derived trypsin, pose significant risks including viral contamination and allergenicity, which hinder their use in food production. To overcome this challenge, we propose a food regulatory pre-check framework, which prioritizes regulatory compliance before functional testing. Following this framework, we developed iDisper, a novel food-grade animal-free cell dissociation reagent composed of approved food additives (papain and trisodium citrate). Specifically, an optimized formulation containing 2 mg/mL papain demonstrated a dissociation efficacy and viability comparable to those of commercial agents in primary avian cells. Furthermore, iDisper demonstrated 8-month stability at −20 °C. This study validated the proposed framework as an effective strategy for developing safe and practical reagents for the cell-based food industry.

## Introduction

For commercially viable cell-based products, the entire production process must follow the same rigorous safety standards as conventional foods [1–3]. In the coming decade, compliance with established food safety management systems, such as HACCP and FSSC 22000, will likely be required [4]. These standards apply not only to final products but also to all raw materials including cell-based food. Therefore, comprehensive reassessment of manufacturing processes is essential. Recent reports on cell-based product safety emphasize that strictly controlling and characterizing all production inputs —including the composition of cell manipulation reagents— is a critical challenge for regulatory compliance [5,6]. This necessitates a fundamental shift in development philosophy from the earliest stages of product design; however, such a comprehensive approach has yet to be widely adopted in practice.

Cell dissociation is essential for both the initial cell isolation and subsequent passaging to expand the cell population [7]. However, trypsin, a commonly used cell dissociation agent, is typically derived from porcine sources. Its animal origin poses

**Data availability statement:** All relevant data are within the manuscript and its Supporting Information files.

**Funding:** This study was supported by the Small Business Innovation Research Program of the Ministry of Agriculture, Forestry and Fisheries of Japan under the "Demonstration of a Production System for Cellular Foods Utilizing CulNet Supernatant" (N020) project. The funders had no role in the study design, data collection and analysis, decision to publish, or preparation of the manuscript.

**Competing interests:** All authors are employees of IntegriCulture Inc. The company developed, and has a commercial interest in, the product described in this manuscript. The authors declare this affiliation as a potential conflict of interest. This does not alter our adherence to PLOS ONE policies on sharing data and materials.

inherent contamination risks for food applications [8,9]. Alternatives, such as thermo-responsive polymers [10,11] and other commercial reagents, also face regulatory barriers. As exemplified by Japan's Food Sanitation Act, many countries strictly control the use of substances through a positive list system [12]. Research-stage synthetic polymers are typically excluded from these lists, and agents with undisclosed compositions cannot be used in food manufacturing.

Regulatory scrutiny of animal-derived ingredients is a key hurdle in cell-based food production. Public FDA filings show that manufacturers consistently highlight the risk of viral contamination from porcine trypsin, often requiring additional safety validation [13–17]. Thus, whether from allergens or viruses, the use of animal-derived ingredients is a key consideration in cell-based food production.

Therefore, a dedicated upstream development process is crucial to integrate safe ingredient selection and consistent product definitions. In this study, an "upstream development process" is defined as a framework that extends beyond simply identifying functional ingredients to include: (1) assessing compliance with food regulations, (2) verifying performance, and (3) evaluating the industrial feasibility as the agent for this field. To demonstrate the effectiveness of this framework, its value lies not only in avoiding animal-derived components but also in selecting ingredients with well-characterized, manageable risks within established food safety paradigms.

As a model case, we developed a cell dissociation agent using food-grade ingredients, including papain, with a known allergen profile, including papain. This approach intentionally shifts the challenge of managing the unknown risks of non-food-grade or composition-undisclosed reagents to a known, visible hazard, providing a more transparent path to food safety. The effectiveness of this framework was demonstrated by selecting ingredients in accordance with food regulations and validating the agent's performance against commercial alternatives. By establishing this foundational perspective, this study not only identifies future industrialization challenges but also paves the way for optimizing the framework through quantitative assessments of supply chain stability, costs, and relative risk.

## Materials and methods

### iDisper reagent preparation

The iDisper was prepared by mixing food-grade papain (W-40; Amano Enzyme, Aichi, Japan), trisodium citrate (Marugo Corporation, Saitama, Japan), and potassium chloride (Kanto Chemical, Tokyo, Japan) in purified water at final concentrations of 2, 0.37, and 28.72 mg/mL, respectively. The reagents were weighed using a precision electronic balance (FX-5000i; A&D, Tokyo, Japan) and added sequentially to water. To ensure complete dissolution, the mixture was stirred at 550 rpm using a magnetic stirrer for 1 h at room temperature (25°C). The solution was sterilized using a Nalgene™ Rapid-Flow™ Sterile Disposable Filter Unit (0.2 μm pore size; 431433; Thermo Scientific, MA, USA). iDisper was stored at −20 °C until use. Before use, the solution was thawed and heated to 37 °C in a water bath. The physicochemical properties were verified using a pH meter (FP20; Mettler Toledo, Switzerland) and an osmometer (Osmostat OM-6040; ARKRAY, Tokyo, Japan). The pH and osmolality

were measured to ensure they fell within the ranges of 5.5–6.0 and 295±5 mOsm/kg, respectively, without requiring any further adjustment. To validate sterility, the filtered solution was plated on Soybean-Casein Digest (SCD) agar and incubated at 35°C. Colony-forming units (CFU) were quantified using an automatic colony counter (Scan500; Interscience, Saint Nom la Bretêche, France). Internal quality control records confirmed 0 CFU across independent manufacturing lots. iDisper was stored at −20 °C until use.

## Cell source, cell culture and evaluation method

Primary cells were isolated from the duck liver and chicken muscle. Fertilized duck eggs were purchased from Shiina Hatchery (Japan), and fertilized chicken eggs were obtained from Yamagishi (Wakayama, Japan). Both types of eggs (containing embryo tissue) were incubated in a P-008 incubator (Showa Furanki, Saitama, Japan) maintained at 37 °C and 60% humidity. The incubation periods were set to 14 and 12 days for the duck and chicken eggs, respectively, to ensure appropriate embryonic development. To ensure sterility, the egg surface was sterilized with ethanol, followed by immersion in sodium hypochlorite solution (Kitchen Haiter; Kao, Japan) for 30 min. Embryos were aseptically extracted in a laminar flow hood, and liver tissues were dissected. The tissues were mechanically dissociated in I-MEM without enzymes using a gentleMACS™ Dissociator (Miltenyi Biotec, Germany). The cell suspension was centrifuged at 300×g for 4 min at room temperature, and the pellet derived from the two embryos was pooled and seeded into a single T175 flask to minimize individual variability. The chicken embryonic fibroblast cell line DF-1 (CRL-12203) was obtained from ATCC.

The cells were maintained and cultured under standard conditions (37°C, 5% CO2) in a basal medium, I-MEM [18], supplemented with 10% Yolk Lipoprotein (YLP), an egg-derived serum substitute [19]. Before cell seeding, the culture vessels were coated with the gelatin-based coating agent iCoater, as described in our previous study [20]. For each passage in the cell-expansion regimen, cells were initially seeded at 25% confluency ($5 \times 10^6$ cells/T175 flask) based on these counts and cultured until an approximate 4-fold expansion reached approximately 100% confluency ($20 \times 10^6$ cells/ T175 flask), at which point they were detached using iDisper. The detached cells were then recovered by centrifugation at 300×g for 5 min at room temperature. For specific experiments to determine cell doubling times, cells were cultured and analyzed as they reached approximately 70% confluency.

Cell dissociation efficacy was evaluated using 12-well plates. After removing the culture medium from each well and washing once with PBS (D-PBS(-) without Ca and Mg; 048−29805; FUJIFILM Wako, Tokyo, Japan), 1 mL of either prewarmed iDisper or a control reagent (TrypLE™ Express; 12605010; Thermo Fisher Scientific, MA, USA) was added. The plates were incubated for 5 min, after which the cells were detached by pipetting. The resulting cell suspension was collected and centrifuged at 300×g for 5 min at room temperature, and the cell number and viability were measured using an automated cell counter (NucleoCounter® NC-202™; ChemoMetec, Denmark) with Via2-Cassette (941−0024; ChemoMetec, Denmark) containing acridine orange and 4′,6-diamidino-2-phenylindole (DAPI) fluorescent dyes, according to the manufacturer's instructions. The cell morphology before and after dissociation was observed using a phase-contrast microscope (BZ-X810; Keyence, Tokyo, Japan).

Primary duck-liver-derived cells were seeded in 6-well plates. For the assay, the culture medium was aspirated, and the wells were washed once with 2 mL of I-MEM. Subsequently, iDisper solutions (0.5 mL) containing papain at different concentrations (1, 2, or 4 mg/mL) or the control reagent TrypLE Express were added to each well. Plates were incubated at 37°C, and cell detachment was monitored and imaged at 0, 5, and 10 min using a phase-contrast microscope. After the final point, I-MEM (2.5 mL) was added, and the cells were thoroughly suspended by pipetting. The cell suspension was transferred to a 15 mL tube and centrifuged at 300×g for 3 min at room temperature. The supernatant was removed, and the cell pellet was resuspended in 1 mL of PBS. The live cell number and particle size distribution were determined using the Cell Counter model R1 (Olympus, Tokyo, Japan) in S1, S2 Figs. For proliferation analysis
(S3 Fig), cell confluence (percentage of occupied area) was monitored using the IncuCyte® SX5 Live-Cell Analysis System (Sartorius, Germany). Time-lapse images were captured, and confluence was quantified using the system's integrated software.

 

### RNA extraction and sequencing

Total RNA was extracted from cell samples of Anas platyrhynchos domestica at various stages of culture using a standard RNA extraction kit, following the manufacturer's protocol. RNA concentration and purity were assessed using spectrophotometry. After quality assessment, the total RNA samples were sent to Rhelixa (Chiba, Japan) for library preparation and sequencing. Strand-specific libraries were prepared using the NEBNext® Poly(A) mRNA Magnetic Isolation Module and the NEBNext® Ultra™ II Directional RNA Library Prep Kit (Illumina, US). Sequencing was performed using an Illumina NovaSeq 6000 platform (Illumina, US), generating 150 bp paired end reads.

### Bioinformatic analysis

Bioinformatic analysis was conducted using Rhelixa. First, the quality of the raw sequence reads was assessed using FastQC (v0.11.7). Adapter sequences and low-quality bases were then trimmed using Trimmomatic (v0.38). The trimmed reads were subsequently aligned to the *A. platyrhynchos domesticus* reference genome (ZJU1.0, GCF_015476345.1) using HISAT2 (v2.1.0). Gene expression levels were quantified from the mapped reads using featureCounts software (v1.6.3). Raw read counts were normalized and expressed as transcripts per million (TPM) to enable comparisons across samples. The expression profiles of specific genes of interest, such as albumin (ALB) and hepatocyte nuclear factor 4α (HNF4A), were visualized from these TPM values. Differentially expressed genes (DEGs) between sample groups were identified using DESeq2 (v1.24.0).

### Mycoplasma detection

To ensure sterility, mycoplasma contamination was assessed in duck liver-derived cells (passages 2 and 5) propagated using the iDisper. Genomic DNA was extracted using the Dneasy® Blood & Tissue Kit (69504; Qiagen, Germany) according to the manufacturer's instructions. PCR amplification was performed using the e-Myco™ plus Mycoplasma PCR Detection Kit (25237; iNtRON Biotechnology, Korea). The amplification products were visualized using gel electrophoresis to confirm the absence of mycoplasma bands (S4 Fig).

### Statistical analysis

All data are expressed as the mean ± standard deviation (SD) from at least three independent biological replicates. Statistical significance was assessed using the Student's t-test for two-group comparisons or one-way ANOVA with a Bonferroni post-hoc test for multiple comparisons. Statistical analyses were performed using Python with the StatsModel library.

## Results and discussion

### A food regulatory pre-check framework for upstream development in the cell-based food industry

To commercialize the cell-based food industry, it is essential to develop industrial products that meet strict food safety standards. However, because the industry is still in its infancy, there is generally no concept of a development process, particularly upstream development, that includes the industry's unique characteristics. Therefore, we proposed a new upstream development framework "Food Regulatory Pre-check" for cell-based food ingredients (Fig 1). The conventional upstream development process typically involves selecting raw ingredients based on their functionality (Step 1) and then immediately preceding to a performance evaluation (Step 3). However, as with trypsin, this approach risks the conclusion that an ingredient performs well but is unsuitable as food.

A notable feature of our framework is that it prioritizes incorporating "food regulatory compliance assessment" as Step 2. In this step, raw ingredients are rigorously screened for approval as "food" based on criteria set out in Japan's Food Sanitation Act (Table 1). Consequently, many existing options, such as trypsin (animal-derived), temperature-responsive polymers (not listed), and EDTA (limited use), can be eliminated at this stage. In contrast, trisodium citrate (INS No. 331(iii)) and

**Step 1.
Concept Planning**

Functional
Requirements
Definition

**Action:**
Define the market need
based on customer pain
points (e.g., high cost,
safety concerns).

**Outcome:**
A defined business case
and value proposition for
a new product.

**Step 2.
Ingredient Screening**

Functional
Candidate
Search

**Action:**
Create a broad list of
biofunctionally potential
ingredients.

**Candidates:**
Proteases (trypsin,
papain), Chelating
agents (EDTA, citrate)

Food Regulation
Pre-Check

**Action:**
Screen candidates against
key food-grade criteria.

**Key Screening Criteria:**
· On Food Positive List
· Animal-Origin-Free
· Characterized Risk
  Profile (e.g., allergenicity)

**Outcome:**
Selection of viable
candidates (e.g., Papain,
Trisodium Citrate)

**Step 3.
Feasibility Test**

Performance Evaluation

**Action:**
Validate the selected candidates
against the key performance and
feasibility metrics.

**Key Evaluation Metrics:**
· Core Efficacy (vs. standard)
· Versatility (cell types)
· Impact on Cell Health
  (viability, phenotype)
· Long-Term Stability

**Outcome:**
 Identification of a lead candidate
that is both regulatorily compliant
and functionally effective.

**Next Step**

Product Profile
Definition

**Action:**
Synthesize the
performance data from
Step 3 with industrial
feasibility factors to make a
final go/no-go decision.

**Key Assessment Criteria:**
 · Validated Performance
  (from Step 3)
 · Manufacturing Cost
 · Scope of application
 · Supply Chain Stability

**Outcome:**
Final approval of the lead
candidate and transition to
the downstream
development process.

**Fig 1. "Food Regulatory Pre-Check" upstream development framework.** The proposed four-step process begins with (1) Concept Planning, where markets are defined. The core of the framework is (2) Ingredient Screening utilizes the "Food Regulation Pre-Check" (yellow box) as a gatekeeper. This step filters infinite candidates down to a finite compliant list based on global regulations and cultural safety (e.g., Kosher status). Viable candidates then undergo (3) a Feasibility Test, where performance is validated against key metrics. Finally, the lead candidate proceeds to (4) Product Profile Definition, which marks the transition to the downstream development process.

papain (INS No. 1101(ii)), which are listed as food additives in the International Food Standard Codex Alimentarius, are the most reasonable options from the perspective of regulations, including global imports and exports.

The efficiency of this framework is quantitatively supported by the "reduction of search space." Traditional reagent development often requires the screening of vast, effectively infinite libraries of synthetic compounds. In contrast, this framework utilizes the "Positive List" system of food regulations as a beneficial constraint, immediately reducing the candidate pool from an infinite number to a finite, manageable list at the start of development (Day 0). For example, the number of enzyme preparations with established specifications in the WHO JECFA Database [21] is limited to approximately 100 distinct source organisms. This drastic reduction in the screening scope mathematically validates the high efficiency of this framework compared with conventional blind screening.

This framework also addresses practical business and cultural issues. First, checking regulations early ensures that the product complies with global standards. Second, it ensures "cultural safety" by including religious criteria (e.g., Kosher status in Table 1) from the start. Finally, it aids corporate decision-making. In business, it is often difficult to stop a project once a budget is assigned for testing (Feasibility Test). By requiring a regulatory "Pre-Check," this framework creates a clear decision point to prevent wasted resources and establishes a formal standard for the industry.

## Performance evaluation of iDisper: Dissociation performance, cell versatility, and long-term stability

The iDisper performance was evaluated using an industrially relevant cell population. We used primary duck liver-derived cells, which are used as raw materials for cell-based food. Under standard culture conditions, these cells proliferated as adherent cells and were predominantly mesenchymal cells. To evaluate the performance of the pre-culture step on cell

**Table 1. Multi-faceted evaluation of candidate cell dissociation technologies based on food safety and manufacturing principles.**

| Category | Ingredients | Origin/ Overview | Cell Culture Precedent | Codex Number | Singapore | USA | Israel | Japan | Allergen |
|---|---|---|---|---|---|---|---|---|---|
| **Plant-Derived Enzymes** | Papain # | **Papaya-derived** | ○ | ○ (INS No. 1101(ii)) | ○ | ○ (21 CFR 184.1585) | ○ | ○ | × |
| | Actinidin | Kiwi-derived | × | × | ○ | ○ | ○ | ○ | × |
| | Ficin | Fig-derived | × | ○ (INS No. 1101(iv)) | ○ | ○ (21 CFR 184.1316) | ○ | ○ | × |
| **Animal-Derived Enzymes** | Trypsin | Porcine pancreas-derived | ○ | × | ○ | ○ (21 CFR 184.1914 | △ (Non Kosher) | ○ | × |
| | Pepsin | Digestive enzymes from pigs, etc. | × | × | ○ | ○ (21 CFR 184.1595) | △ (Non Kosher) | ○ | × |
| **Chelating Agents** | Trisodium citrate # | **Organic acid found in citrus fruits** | ○ | ○ (INS No. 331(iii)) | ○ | ○ (21 CFR 184.1751) | ○ | ○ | ○ |
| | EDTA | Chemically synthesized | ○ | ○ (INS No. 385) | ○ | ○ (21 CFR 573.360) | △ (Use limit) | △ (Use limit) | ○ |
| **Recombinant Enzymes** | Recombinant Collagenase | Microbial origin. Degrades collagen. Standard reagent for 3D culture dissociation, etc. | ○ | × | × | × | × | × | ○ |
| **Others** | Thermoresponsive polymer | Synthetic polymer, etc. | × | × | × | × | × | × | ○ |

Legend; ○: Proven track record, △: Limited track record, ×: Significant challenges, #: Materials selected in this study.

quality, we compared gene expression profiles in cells at an early stage of passage (P0) and late passages (P2 and P6) (Fig 2). As a result, the expression of ALB and HNF4A, which are hepatocyte maturation markers [22], was significantly decreased in P2 and P6 cells compared to that in P0 cells. In addition, the expression levels of COL3A1 and DCN, which are highly expressed in mesenchymal cells [23], were significantly increased. Hepatocytes generally have a low proliferation profile [24]. This suggests that a cell population with a higher ability to adhere to the planar substrate and that is suitable for proliferation may have been selected. This indicated that the characteristics of the cells selected in the standard 2D cell culture were maintained.

Development of iDisper began with trisodium citrate as a gentle baseline chelating agent, considering the delicacy of primary cells [25]. However, when tested on primary duck liver-derived cells, citrate alone failed to achieve complete detachment, as shown by microscopy and low cell counts (Fig 3a-3c). Particle size analysis indicated the presence of large aggregates with citrate alone (S1 Fig). To improve efficacy, we investigated the addition of food-grade protease papain. A comparative evaluation demonstrated that the combination of trisodium citrate and papain significantly improved cell dissociation. A papain concentration of 2 mg/mL yielded a dissociated cell number and a single-cell particle size distribution comparable to the commercial reagent TrypLE™ Express (S2 Fig). In contrast, 1 mg/mL was insufficient for effective dissociation. Although a concentration of 4 mg/mL also induced detachment, it caused a significant delay in subsequent cell proliferation compared to 2 mg/mL (S3 Fig). Although papain alone was effective, the iDisper formulation was finalized by combining papain with trisodium citrate. This design was chosen to mimic the dual mechanism of standard trypsin-EDTA reagents, utilizing proteolysis (2 mg/mL papain) and chelation (citrate) to target distinct cell adhesion factors.

Furthermore, iDisper consistently supported robust expansion through serial passaging, simulating a cell-based food production line (Fig 3d-3e). However, the 4.1% failure rate at passage 0 indicates a limitation of iDisper in overcoming the inherent variability in primary avian cells. This instability at the initial isolation stage acts as a bottleneck, potentially

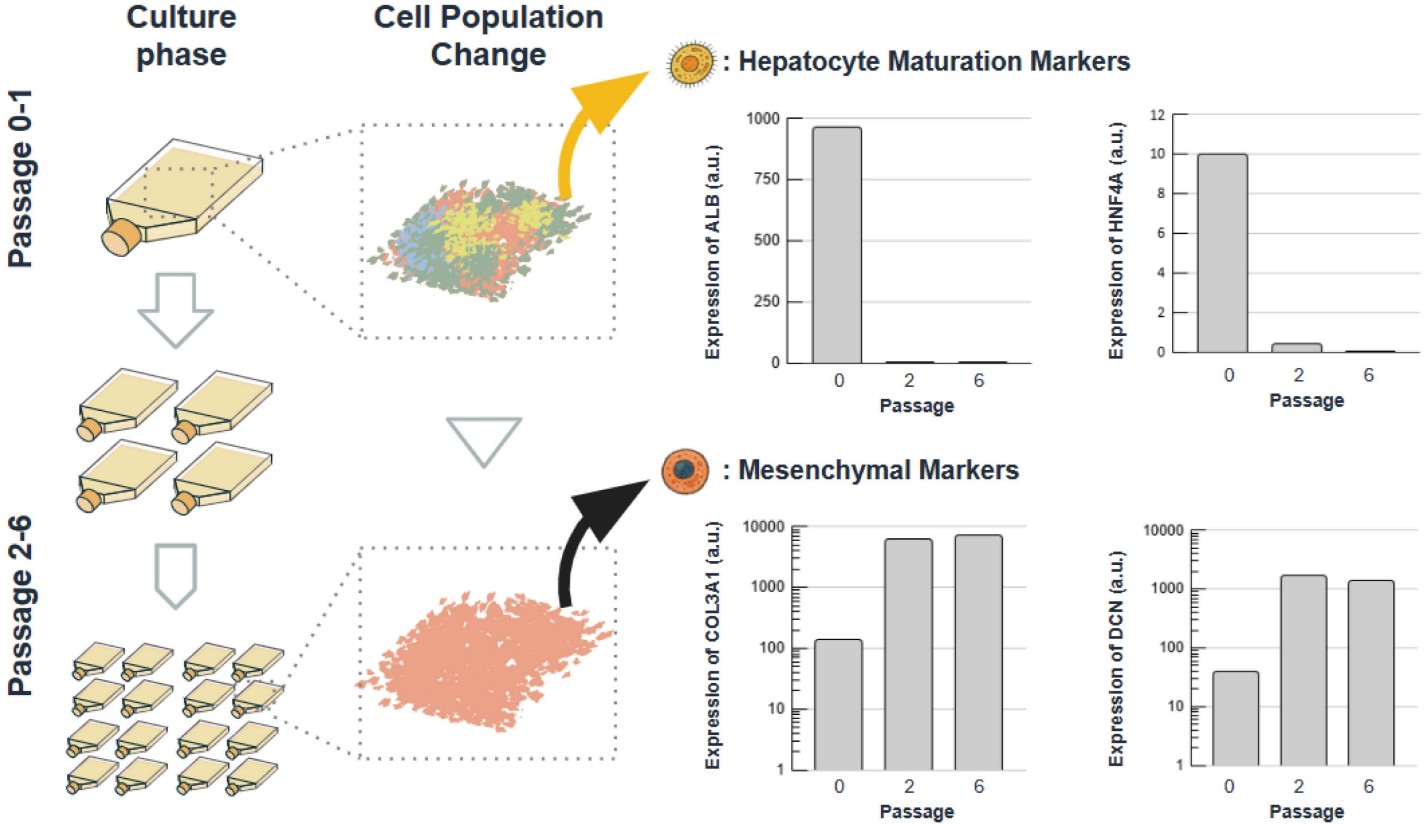

**Fig 2. Gene expression changes during serial passage.** Gene expression in duck liver-derived cells was analyzed at passage 0 (P0), P2, and P6. Expression of hepatocyte maturation markers ALB and HNF4A decreased significantly by P2. Conversely, expression of mesenchymal markers COL3A1 and DCN, associated with proliferation, increased significantly by P2. These changes suggest the selection of a more proliferative cell population during culture expansion. Gene expression values are shown in arbitrary units (a.u.).

reducing overall production efficiency. Therefore, optimizing the iDisper for robust primary isolation while being effective for expansion remains a critical challenge.

Mycoplasma testing confirmed that the cells passaged with different lots of iDisper (P2 and P5) were negative for contamination (S4 Fig).

To evaluate its broader utility, we tested the iDisper on other avian cell types, including primary chicken muscle cells and DF-1. The iDisper achieved robust cell numbers and high viability across all cell types (Fig 4a). The product showed excellent storage stability, with no performance loss after 8 months at −20 °C (Fig 4b). The detachment performance decreased to approximately 30% after 12 months (S5 Fig). However, a limitation of the current study is that the dissociation efficacy of iDisper was exclusively validated using avian cells. Given that bovine and porcine species are next targets for the cell-based food industry, future studies should validate this versatility across a broad range of industrially relevant species, including mammalian and aquatic cells.

We compared iDisper to the animal-free recombinant enzyme TrypLE™ Express to highlight two viable paths for the food industry: the food-grade approach and the recombinant protein approach. Our results demonstrated that iDisper was robust and effective, thus validating the food-grade path for scalable expansion.

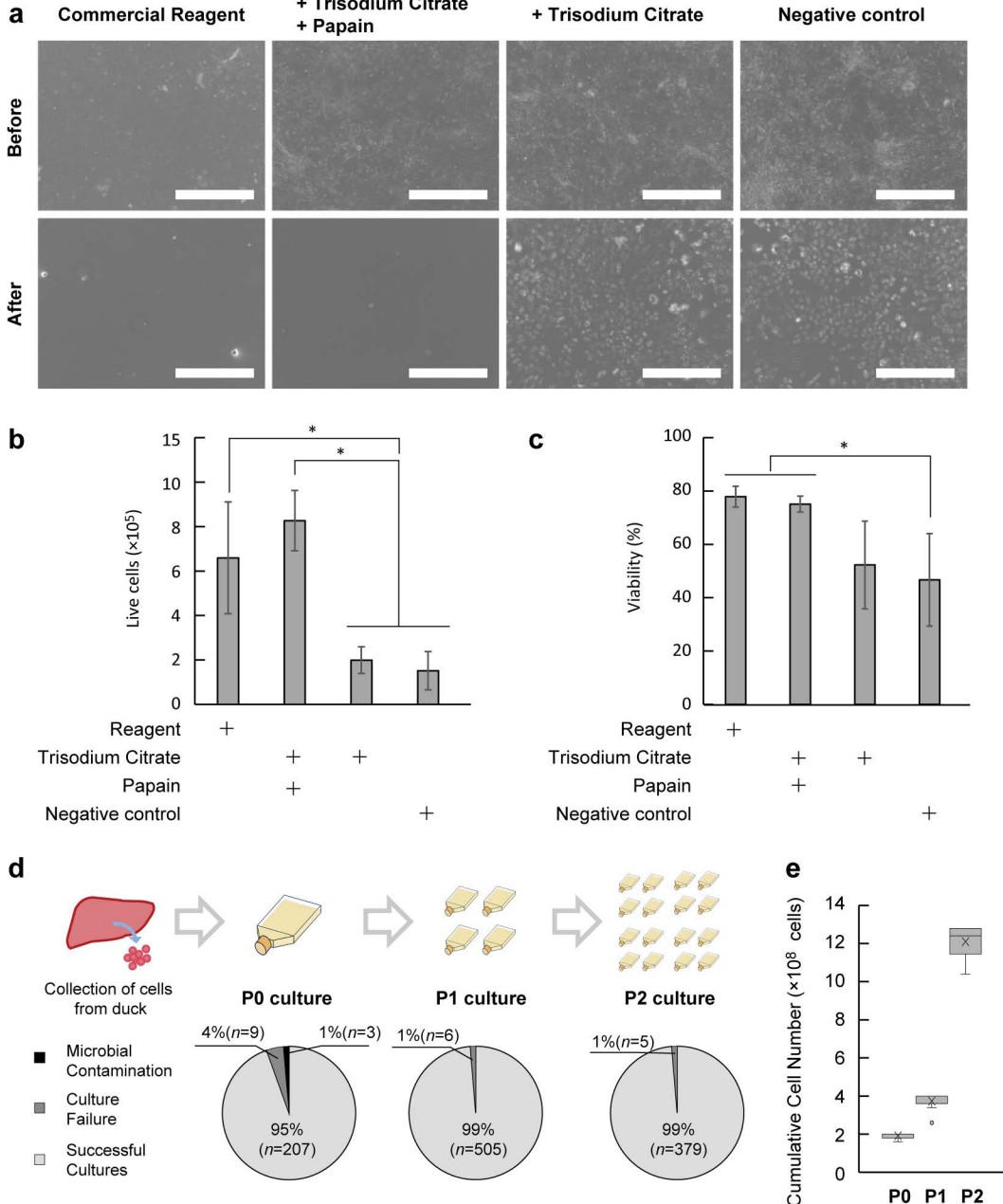

**Fig 3. Performance of iDisper on primary duck liver-derived cells. (a)** Representative brightfield images showing cell detachment. The iDisper (+ Trisodium Citrate + Papain) condition shows detachment comparable to the Reagent, whereas Trisodium Citrate alone (Negative control) showed incomplete detachment. Scale bar = 0.5 mm. **(b)** Live cells recovered and (c) cell viability after detachment for each condition. Data are presented as mean ± SD ($n = 3$, * $p < 0.05$). **(d)** Success rates for cell expansion from passage 0 (P0) to P2. Pie charts show the proportion of successful cultures versus those lost to culture failure or microbial contamination. The percentages of culture outcomes were calculated by dividing the number of specific events by the total number of evaluated cultures ($n$) for each passage; the total sample sizes are $n = 219$ for P0 (comprising 207 successful, 9 contaminations, and 3 failures), $n = 511$ for P1 (505 successful, and 6 failures), and $n = 384$ for P2 (379 successful, and 5 failures). **(e)** Cumulative cell number increase from P0 to P2. Box plots show the distribution of total cells for cultures that met pre-defined criteria for subsequent production stages.

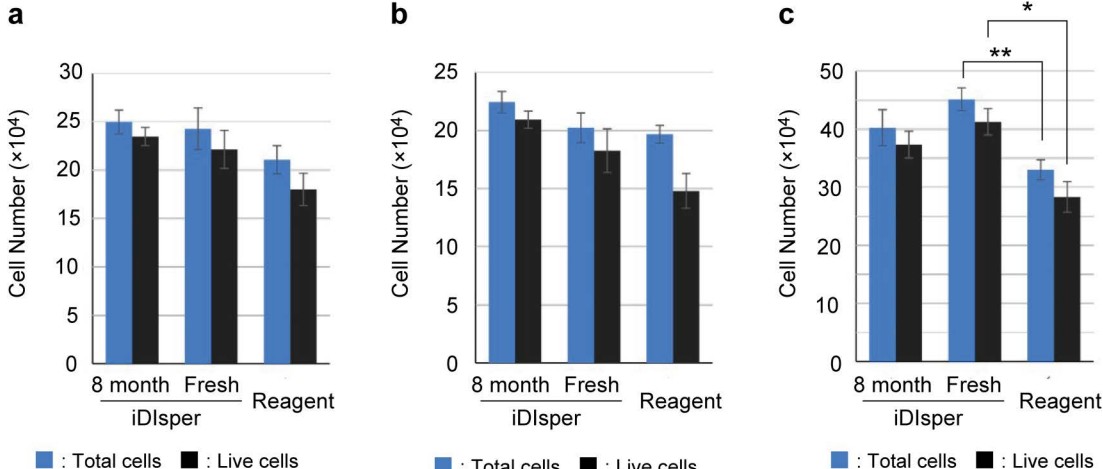

**Fig 4. Versatility and stability of iDisper across different cell types.** Performance of freshly prepared iDisper ("Fresh"), iDisper stored for 8 months at −20 °C ("8 month"), and a commercial control ("Reagent") was compared on duck liver-derived cells **(a)**, chicken muscle-derived cells **(b)**, and the DF-1 cell line **(c)**. Blue and black bars represent total and live cell numbers, respectively. Data are presented as mean±SD ($n=3$). Statistical analyses were performed separately for total and live cell counts using Student's t-test followed by Bonferroni correction. * $p<0.05$, ** $p<0.01$ compared to the Reagent group.

## Industrial production of food-grade cell culture materials and future challenges

The upstream development process proposed in this study must be integrated with the downstream development process to bring products to market. In a previous study, we established a heat-based sterilization process [20]. However, as iDisper's core component, papain, is heat-sensitive, an industrial-scale product is likely to be a powder. This is a common formulation for food ingredients such as seasonings or powdered soup bases. Future developments must address key manufacturing parameters for powders, such as controlling the particle size to ensure product uniformity, solubility, and stability [26,27]. Currently, the specific limits of scalability for this powder formulation remain unquantified. Moving beyond laboratory-scale proof-of-concept will necessitate rigorous quantitative evaluations to define the maximum viable batch sizes and the precise moisture thresholds acceptable for long-term storage.

The iDisper is transparently treated as a dispersion with defined allergen characteristics, enabling its management within food safety framework. This directly contrasts with agents with secret compositions, which pose "hidden risks," make appropriate food safety assessments difficult, and contribute to serious issues such as food allergen recalls in the conventional food industry [28].

By contrast, our framework yields products with visible, manageable risks. The complete composition and our proposed labeling, conceptually based on Japanese food labeling standards, are shown in (Supporting Information; S1 Table). When compared to recent FDA pre-market consultations for cell-cultured meat [13–17]—which rigorously evaluate whether allergenic processing aids remain in the final product—our approach similarly ensures safety by transparently declaring known allergens (e.g., "Contains enzymes derived from papaya") A logical consequence is that the final cell-based food product may require allergen labeling to ensure consumer safety and choice, which is a standard and established practice in the food industry [29]. The core contribution of this study is to provide transparent starting materials that make allergen risk manageable, shifting the industry's approach from dealing with unknown variables to overcoming known ones. Transparency is an essential step for downstream producers to validate allergen removal or inactivation processes. For instance, a practical manufacturing process would likely require validated mitigation steps, such as post-dissociation washing steps or terminal heat treatment to inactivate residual enzymes. In terms of temporal stability, the allergen remained structurally

robust throughout the process. Our data (Fig 4b) showed stability for 8 months at −20°C, and Vicente et al. [30] reported that papain retains its stability with a half-life of 10 days in aqueous media. This quantitative evidence indicates that the allergen does not degrade naturally during cell dissociation, necessitating active removal strategies.

## Conclusions

This study proposed and validated a food regulatory pre-check framework for developing safe, practical, and animal-free materials for the cell-based food industry. The resulting reagent, iDisper, demonstrated dissociation efficacy comparable to that of commercial alternatives and excellent long-term stability. This study shifts the paradigm from managing unknown risks in conventional reagents to addressing the visible and manageable risks of transparently formulated food-grade materials. Future studies should optimize industrial production and validate allergen mitigation strategies.

## Supporting information

**S1 Fig. Particle size distribution of duck liver-derived cells immediately after detachment.** Green and red bars indicate live and dead cells, respectively. The brightfield images below each graph show the cell suspension used for the measurement. Observationally, the particle size distribution for the iDisper condition (+ Trisodium Citrate + Papain) showed a primary peak between 10–15 µm, similar to the Commercial Reagent. In contrast, the Trisodium Citrate alone and the Negative control conditions showed a main peak closer to 10 µm with a broader distribution, suggesting the presence of cell aggregates. These data are presented as a qualitative reference.
(TIF)

**S2 Fig. Optimization of papain concentration for effective cell dissociation.** Phase-contrast microscopy images of primary duck liver cells before treatment (Before) and at 10 min after (After) the addition of iDisper containing papain at 1, 2, or 4 mg/mL. The control reagent, TrypLE™ Express, was treated for 5 min. The "After pipetting" column shows the plate surface after cell collection. All cultures were confluent before treatment. The bar chart shows the number of live cells recovered after detachment. Data are presented as mean ± SD ($n = 3$). Scale bar = 1 mm.
(TIF)

**S3 Fig. Post-detachment proliferation of cells treated with different papain concentrations.** Growth curves of primary duck liver cells over approximately 7 days following detachment with iDisper containing papain at 1, 2, or 4 mg/mL, or with the control reagent TrypLE™ Express. Cell proliferation was evaluated by measuring cell confluence (%) over time using an IncuCyte® live-cell imaging and analysis system. Data are presented as mean ± SE ($n = 3$).
(TIF)

**S4 Fig. Validation of sterility: Mycoplasma PCR test.** Duck liver cells passaged with iDisper were tested using the e-Myco™ plus Kit across three independent lots (Lots A, B, and C). Lanes: Marker: DNA ladder; Positive Control (kit): Positive control showing specific target bands (including 270 bp); Duck liver-derived cell (P2, Lot A): Cells at Passage 2 ($n = 3$); Duck liver-derived cell (P2, Lot B): Cells at Passage 2 ($n = 3$); Duck liver-derived cell (P5, Lot C): Cells at Passage 5 ($n = 2$); Water: Negative control. The presence of internal control bands in all sample lanes confirms successful PCR amplification, while the absence of the specific 270 bp band confirms that all cell cultures were free of mycoplasma contamination.
(TIF)

**S5 Fig. Evaluation of iDisper stability after long-term storage.** The cell detachment performance of iDisper stored for 12 months at - 20°C was compared against that of a commercial reagent. Data are presented as mean ± SD ($n = 3$). The cell recovery rate after 12 months was below 50% of the control, which did not meet our internal criteria for further statistical analysis.
(TIF)

**S1 Table. Proposed food label for iDisper designed to declare its 'manageable risk'.** This label demonstrates our framework for avoiding unknown risks in favor of managing known risks. As established in our previous study [20], classifying products as food and providing transparent information is fundamental for reliable safety management in cellular agriculture.
(DOCX)

**S1 File. Raw data for Fig2–4'.**
(ZIP)

**S2 File. Raw data for S1-5 Fig.**
(ZIP)

## Acknowledgments

The authors express their sincere gratitude to food regulatory experts and government officials who provided invaluable advice and insights for this study. We also thank all members of the Production Development Department at IntegriCulture Inc. for their helpful discussions

## Author contributions

**Conceptualization:** Hiroaki Hatano.

**Data curation:** Hiroaki Hatano.

**Formal analysis:** Hiroaki Hatano, Misaki Sawada.

**Investigation:** Misaki Sawada, Misaki Kitsukawa, Azusa Hayakawa.

**Methodology:** Hiroaki Hatano, Misaki Sawada.

**Project administration:** Misaki Sawada.

**Supervision:** Hiroaki Kondo, Ikko Kawashima.

**Validation:** Misaki Sawada, Misaki Kitsukawa, Azusa Hayakawa.

**Visualization:** Hiroaki Hatano.

**Writing – original draft:** Hiroaki Hatano.

**Writing – review & editing:** Hiroaki Hatano.

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
