## [Decision Letter · Decision Letter 0]

21 Dec 2025

Dear Dr. Kawashima,

Thank you for submitting your manuscript to PLOS ONE. After careful consideration, we feel that it has merit but does not fully meet PLOS ONE’s publication criteria as it currently stands. Therefore, we invite you to submit a revised version of the manuscript that addresses the points raised during the review process.

A letter that responds to each point raised by the academic editor and reviewer(s). You should upload this letter as a separate file labeled ‘Response to Reviewers’.A marked-up copy of your manuscript that highlights changes made to the original version. You should upload this as a separate file labeled ‘Revised Manuscript with Track Changes’.An unmarked version of your revised paper without tracked changes. You should upload this as a separate file labeled ‘Manuscript’.

If applicable, we recommend that you deposit your laboratory protocols in protocols.io to enhance the reproducibility of your results. Protocols.io assigns your protocol its own identifier (DOI) so that it can be cited independently in the future. For instructions see: https://journals.plos.org/plosone/s/submission-guidelines#loc-laboratory-protocols. Additionally, PLOS ONE offers an option for publishing peer-reviewed Lab Protocol articles, which describe protocols hosted on protocols.io. Read more information on sharing protocols at . Additionally, PLOS ONE offers an option for publishing peer-reviewed Lab Protocol articles, which describe protocols hosted on protocols.io. Read more information on sharing protocols at . Additionally, PLOS ONE offers an option for publishing peer-reviewed Lab Protocol articles, which describe protocols hosted on protocols.io. Read more information on sharing protocols at . Additionally, PLOS ONE offers an option for publishing peer-reviewed Lab Protocol articles, which describe protocols hosted on protocols.io. Read more information on sharing protocols at https://plos.org/protocols?utm_medium=editorial-email&utm_source=authorletters&utm_campaign=protocols..

We look forward to receiving your revised manuscript.

Kind regards,

Shengqian Sun

Academic Editor

PLOS One

1. Please ensure that your manuscript meets PLOS ONE’s style requirements, including those for file naming. The PLOS ONE style templates can be found at

“This research was funded by the Ministry of Agriculture, Forestry and Fisheries of Japan (MAFF) through the Small/Startup Business Innovation Research Program (SBIR), grant number N020.”

“All authors are employees of IntegriCulture Inc. The company developed, and has a commercial interest in, the product described in this manuscript. The authors declare this affiliation as a potential conflict of interest.”

We note that one or more of the authors are employed by a commercial company: name of commercial company.

1.        Please provide an amended Funding Statement declaring this commercial affiliation, as well as a statement regarding the Role of Funders in your study. If the funding organization did not play a role in the study design, data collection and analysis, decision to publish, or preparation of the manuscript and only provided financial support in the form of authors’ salaries and/or research materials, please review your statements relating to the author contributions, and ensure you have specifically and accurately indicated the role(s) that these authors had in your study. You can update author roles in the Author Contributions section of the online submission form.

4. When completing the data availability statement of the submission form, you indicated that you will make your data available on acceptance. We strongly recommend all authors decide on a data sharing plan before acceptance, as the process can be lengthy and hold up publication timelines. Please note that, though access restrictions are acceptable now, your entire data will need to be made freely accessible if your manuscript is accepted for publication. This policy applies to all data except where public deposition would breach compliance with the protocol approved by your research ethics board. If you are unable to adhere to our open data policy, please kindly revise your statement to explain your reasoning and we will seek the editor’s input on an exemption. Please be assured that, once you have provided your new statement, the assessment of your exemption will not hold up the peer review process.

6. We notice that your supplementary table is  included in the manuscript file. Please remove them and upload them with the file type ‘Supporting Information’. Please ensure that each Supporting Information file has a legend listed in the manuscript after the references list.

Reviewers’ comments:

Reviewer’s Responses to Questions

**Comments to the Author**

1. Is the manuscript technically sound, and do the data support the conclusions?

Reviewer #1: Yes

Reviewer #2: Partly

2. Has the statistical analysis been performed appropriately and rigorously?

Reviewer #1: Yes

Reviewer #2: I Don’t Know

3. Have the authors made all data underlying the findings in their manuscript fully available?

Reviewer #1: Yes

Reviewer #2: Yes

4. Is the manuscript presented in an intelligible fashion and written in standard English?

Reviewer #1: Yes

Reviewer #2: Yes

Reviewer #1: The manuscript introduces a groundbreaking framework for pre-checking food regulations in the development of iDisper, a food-grade cell dissociation reagent crafted from papain and trisodium citrate, aimed at enhancing cell-based food production. The discussion highlights significant safety concerns in cellular agriculture, showcasing encouraging initial findings on the effectiveness of dissociation in avian cells. Nevertheless, considerable methodological, analytical, and interpretive concerns require substantial revisions prior to contemplating publication in PLOS ONE.

Significant Issues

Abstract and Language Issues: Grammatical errors (e.g., "Advancement... require" should be "requires"; "demon-strated" hyphenation artifact) and awkward phrasing diminish the overall clarity. Revise the abstract for clarity and rectify any typographical or formatting mistakes present.

The "Food Regulatory Pre-Check" framework (Fig 1, Table 1) presents an intriguing concept; however, it falls short in terms of thorough validation. Present numerical data that contrasts development timelines and costs with traditional methods, and enhance Table 1 by incorporating global regulatory information beyond Japan, such as the FDA GRAS status for papain. Explain why this represents a "new" framework instead of conventional food additive screening.

The methods presented are lacking in detail: The sourcing of cells from duck liver and chicken muscle does not include necessary ethical approvals, provenance information, or controls for variability. The preparation of iDisper lacks specific details regarding pH levels, mixing conditions, and does not provide validation for sterility beyond the filtration process. It is essential to ensure that statistical methods are supported by the deposition of raw data (ChemRxiv DOI provided, but please verify accessibility). Incorporate comprehensive protocols to ensure reproducibility.

The interpretation of results appears exaggerated: Assertions of "comparable" performance to TrypLE Express (Figs 2-3) overlook significant differences in incubation duration, cell types, and metrics for passaging success. Figure 2d pie charts illustrate significant failure rates (approximately 30-50%) that cannot be accounted for by contamination or reagent-related problems. Examine the constraints of testing exclusively on birds and analyze the duration of papain allergens in a quantitative manner.

Concerns regarding conflict of interest and commercial bias arise from the involvement of all authors from IntegriCulture Inc. (the product developer), particularly in relation to provisional labeling (S1 Table). Offer unbiased validation data or external testing; reduce promotional language (e.g., "Everyone Can Make Cell-Based Food").

Reviewer #2: Hatano and colleagues investigate a novel food-grade animal-free cell dissociation reagent, “iDisper”, to achieve efficient dissociation and viability of primary avian cells (duck liver and chicken muscles), that would be used in downstream cell-based food products e.g. lab grown meat. Commonly used cell dissociation agents (e.g. trypsin) poses contamination risks, such as viral contamination or allergens, for food applications. This manuscript highlights a novel dissociation reagent which includes papain (a protease from unripe papaya) and validating the performance of the new dissociation reagent against commercial alternatives. Furthermore, they propose a new upstream development “Food Regulatory Pre-Check” framework for cell-based food ingredients.

• You state that iDisper could maintain passaging of cells with hepatic stellate identity and stable marker expression, however your results S2 Appendix – S1 indicate a loss of ALB and HNF4A in P2 and P6 cells (both hepatocyte maturation markers). Can you please clarify why you conclude that iDisper maintains cells with hepatic stellate identity i.e. what markers are you looking at to make that conclusion?

• You examined microbial contamination in passaged cells, but did you confirm that cell cultures were mycoplasma free?

• There is no discussion of Sup Fig 3 and 4 in the main manuscript – please include discussion especially since you conclude that papain with trisodium citrate is better than papain alone. You have stated that papain alone could not achieve dissociation to the levels of papain with trisodium citrate (line 133), however you currently have not presented/discussed the results to support that claim (or please move your S2 Appendix discussions into the main body of the manuscript). I see that Sup Fig 3 indicates dissociation of duck liver cells with papain alone at various concentrations, and it appears that papain alone did achieve similar dissociation results compared to the TryplE Express reagent. Hence I’m not convinced that papain + trisodium citrate is required over papain alone.

• Fig 3 – no indication of A, B or C on the figure. Also can you indicate statistical significance on the figures.

• Methods section of the main manuscript could be expanded to include S1 Appendix (do not understand why this methods has been moved to the supplementary section.

• When washing the dissociated cells with PBS, is there calcium and magnesium included in the PBS?

• Details in supplemental methods do not match what is written in main manuscript – for example, in S1 Appendix-S3 Cell detachment assay, you state that the Cell Counter model R1 (Olympus) was used to measure total cell number and viability, but in the main manuscript you state that the NucleoCounter NC-202 was used. Please clarify if both equipment were used.

• Proof-read the manuscript before submission, especially punctuation.

• Table 1 format can be improved.

.

Reviewer #1: No

Reviewer #2: No

---

## [Author Response · Author response to Decision Letter 1]

28 Jan 2026

Dear Editor,

Resubmission of Manuscript ID: PONE-D-25-59887

Title: “A Food-Grade, Animal-Free Cell Dissociation Alternative Product Developed via a Food Regulatory Pre-Check Framework”.

Response to Editor and Reviewers

We would like to express our sincere appreciation of the reviewers for their valuable comments and suggestions. We revised the manuscript based on the comments and suggestions provided below. In this response letter, the reviewers’ comments are presented in italics, the authors’ responses are written in regular font with a five-character indent, and the revised text is in blue.

Editor

We sincerely appreciate the reviewer’s constructive and insightful feedback, which has greatly improved the clarity and quality of our manuscript. We have carefully revised our manuscript in response to these comments. We hope that these revisions adequately address the reviewers’ concerns and that the revised manuscript is now suitable for publication in PLOS One. We look forward to hearing from you.

Journal Requirements

1. Please ensure that your manuscript meets PLOS ONE’s style requirements, including those for file naming

Response 1: We appreciate the guidance. We have carefully reviewed the guidelines and revised the manuscript, including file naming conventions and formatting, to fully comply with PLOS ONE’s style requirements.

2. Please state what role the funders took in the study.

Response 2: We have included the specific role of the funders in the Cover Letter (page 2).

Added text: This study was supported by the Small Business Innovation Research Program of the Ministry of Agriculture, Forestry and Fisheries of Japan under the “Demonstration of a Production System for Cellular Foods Utilizing CulNet Supernatant” (N020) project. The funders had no role in the study design, data collection and analysis, decision to publish, or preparation of the manuscript.

3.1. Please provide an amended Funding Statement declaring this commercial affiliation, as well as a statement regarding the Role of Funders in your study.

Response 3.1: We have amended the Funding Statement to clarify the commercial affiliation and role of the funder.

Added text: The funder provided support in the form of salaries for the authors [HH, MS, MK, AH, HK, and IK], but did not have any additional role in the study design, data collection and analysis, decision to publish, or preparation of the manuscript. The specific roles of these authors are articulated in the ‘author contributions’ section.

3.2. Please also provide an updated Competing Interests Statement declaring this commercial affiliation along with any other relevant declarations relating to employment, consultancy, patents, products in development, or marketed products, etc.

Response 3.2: We have updated both the Cover Letter and Competing Interests Statement to explicitly declare commercial affiliation and confirm adherence to journal policies.

Added text: This does not alter our adherence to PLOS ONE policies on sharing data and materials.

4. When completing the data availability statement of the submission form, you indicated that you will make your data available on acceptance. We strongly recommend all authors decide on a data sharing plan before acceptance, as the process can be lengthy and hold up publication timelines. Please note that, though access restrictions are acceptable now, your entire data will need to be made freely accessible if your manuscript is accepted for publication.

Response 4: We acknowledge the policy regarding data availability. We confirm that all data underlying the findings described in this manuscript are fully available without restrictions. The minimal data set underlying the results, including the data in Figures 2 and 3, is provided in the Supporting Information submitted for this revision.

Response 5: The corresponding author (Ikko Kawashima) has a registered ORCID iD [0009-0003-9612-4007]. We ensured that this ID was successfully linked and validated using an Editorial Manager System.

6. We notice that your supplementary table is included in the manuscript file. Please remove them and upload them with the file type ‘Supporting Information’. Please ensure that each Supporting Information file has a legend listed in the manuscript after the references list.

Response 6: We appreciate your guidance on the formatting. We have removed the supplementary table from the main manuscript file and uploaded it as a separate file labeled ‘Supporting Information’ (S1 Table). Additionally, we have ensured that the legends for all the Supporting Information files are correctly listed at the end of the main manuscript, following the References section.

Reviewer: 1

Comment 1: Abstract and Language Issues: Grammatical errors (e.g., "Advancement... require" should be "requires"; "demon-strated" hyphenation artifact) and awkward phrasing diminish the overall clarity. Revise the abstract for clarity and rectify any typographical or formatting mistakes present.

Response 1: We apologize for grammatical errors and awkward phrasing in the original submission. We have thoroughly revised the manuscript, including the abstract, and have had the text proofread by Cactus Communications, Inc. (Tokyo, Japan). Specific corrections, such as "Advancement... requires", and “demonstrated” have been made.

Comment 2-1: The "Food Regulatory Pre-Check" framework (Fig 1, Table 1) presents an intriguing concept; however, it falls short in terms of thorough validation. Present numerical data that contrasts development timelines and costs with traditional methods, and enhance Table 1 by incorporating global regulatory information beyond Japan, such as the FDA GRAS status for papain. Explain why this represents a "new" framework instead of conventional food additive screening.

Response 2-1: We appreciate the reviewer’s suggestion. Although specific financial data and timelines are proprietary and difficult to benchmark owing to the nascency of the industry, we quantitatively validated the framework’s efficiency by demonstrating the "Reduction of Search Space." Unlike traditional screening of effectively infinite chemical libraries, our framework restricts the candidate pool to a finite "Positive List" (e.g., 104 enzyme preparations in the WHO JECFA Database; verified by "Enzyme Preparation" in the JECFA Online Database on January 16, 2026;

https://apps.who.int/food-additives-contaminants-jecfa-database/) from Day 0. This mathematical constraint to have a significantly higher screening efficiency than conventional methods. We have incorporated this rationale into the Discussion section. Additionally, we updated Table 1 to include regulatory information from Singapore (first approval in 2020), the USA (approval in 2023), and Israel (approval in 2024), which currently lead the global cellular agriculture sector.

Revised text (Page 8, Line 168): The efficiency of this framework is quantitatively supported by the "reduction of search space." Traditional reagent development often requires the screening of vast, effectively infinite libraries of synthetic compounds. In contrast, this framework utilizes the "Positive List" system of food regulations as a beneficial constraint, immediately reducing the candidate pool from an infinite number to a finite, manageable list at the start of development (Day 0). For example, the number of enzyme preparations with established specifications in the WHO JECFA Database [19] is limited to approximately 100 distinct source organisms. This drastic reduction in the screening scope mathematically validates the high efficiency of this framework compared with conventional blind screening.

Comment 2-2: Explain why this represents a "new" framework instead of conventional food additive screening.

Response 2-1: We appreciate the reviewer’s comment. The novelty of this framework is that we reversed the screening order. Instead of just looking for "what works," we prioritize "what is allowed" to address three practical realities:

● Global Rules: Conventional methods often identify high-performing ingredients that are banned in some countries. Our method prevents this by checking global regulations first.

● Cultural Safety: Food is a culture. As shown by the inclusion of Kosher status (Table 1), we checked religious and cultural suitability early to avoid ethical conflicts.

● Business Reality: Many companies find it difficult to stop a project once the budget is approved (Step 3). By setting a mandatory "Pre-Check" (Step 2), we create a clear "Stop/Go" point before spending money. Making this common-sense management rule a formal framework is a novel contribution.

Revised text (Page 9, Line 176): This framework also addresses practical business and cultural issues. First, checking regulations early ensures that the product complies with global standards. Second, it ensures "cultural safety" by including religious criteria (e.g., Kosher status in Table 1) from the start. Finally, it aids corporate decision-making. In business, it is often difficult to stop a project once a budget is assigned for testing (Feasibility Test). By requiring a regulatory "Pre-Check," this framework creates a clear decision point to prevent wasted resources and establishes a formal standard for the industry.

Revised text (Page 9, Line 183): Fig 1. "Food Regulatory Pre-Check" upstream development framework. The proposed four-step process begins with (1) Concept Planning, where markets are defined. The core of the framework is (2) Ingredient Screening utilizes the "Food Regulation Pre-Check" (yellow box) as a gatekeeper. This step filters infinite candidates down to a finite compliant list based on global regulations and cultural safety (e.g., Kosher status). Viable candidates then undergo (3) a Feasibility Test, where performance is validated against key metrics. Finally, the lead candidate proceeds to (4) Product Profile Definition, which marks the transition to the downstream development process.

Comment 3: The methods presented are lacking in detail: The sourcing of cells from duck liver and chicken muscle does not include necessary ethical approvals, provenance information, or controls for variability. The preparation of iDisper lacks specific details regarding pH levels, mixing conditions, and does not provide validation for sterility beyond the filtration process. It is essential to ensure that statistical methods are supported by the deposition of raw data (ChemRxiv DOI provided, but please verify accessibility). Incorporate comprehensive protocols to ensure reproducibility.

Response 3: We appreciate the reviewer’s guidance on improving the transparency and reproducibility of our methods. We have addressed each point as follows:

Ethical Information: We added a specific statement regarding ethical compliance to the manuscript.

Revised text (Page 16, Line 295): Compliance with Ethical Standards

In this study, we used duck eggs embryonated before hatching. According to Japanese institutional guidelines and the SCAW Category A classification, these embryos are not defined as experimental animals; therefore, formal approval from the Institutional Animal Care and Use Committee was not required. All procedures were performed in accordance with the principles of animal welfare to minimize potential distress.

Provenance Information of duck liver and chicken muscle cells: We have expanded the "Materials and Methods" section to include detailed preparation conditions and quality control specifications to ensure reproducibility.

Revised text (Page 4, Line 77): Cell Source, Cell Culture and Evaluation Method

Primary cells were isolated from the duck liver and chicken muscle. Fertilized duck eggs were purchased from Shiina Hatchery (Japan), and fertilized chicken eggs were obtained from Yamagishi (Wakayama, Japan). Both types of eggs were incubated in a P-008 incubator (Showa Furanki, Saitama, Japan) maintained at 37 °C and 60% humidity. The incubation periods were set to 14 and 12 days for the duck and chicken eggs, respectively, to ensure appropriate embryonic development. To ensure sterility, the egg surface was sterilized with ethanol, followed by immersion in sodium hypochlorite solution (Kitchen Haiter; Kao, Japan) for 30 min. Embryos were aseptically extracted in a laminar flow hood, and liver tissues were dissected. The tissues were mechanically dissociated in I-MEM without enzymes using a gentleMACS™ Dissociator (Miltenyi Biotec, Germany). The cell suspension was centrifuged, and the pellet derived from the two embryos was pooled and seeded into a single T175 flask to minimize individual variability.

iDisper Preparation and Sterility Validation: We expanded the "Materials and Methods" section to include detailed preparation conditions and quality control specifications to ensure reproducibility.

Revised text (Page 4, Line 62): The iDisper was prepared by mixing food-grade papain (W-40; Amano Enzyme, Aichi, japan), trisodium citrate (Marugo Corporation, Saitama, Japan), and potassium chloride (Kanto Chemical, Tokyo, Japan) in purified water at final concentrations of 2, 0.37, and 28.72 mg/mL, respectively. The reagents were weighed using a precision electronic balance (FX-5000i; A&D, Tokyo, Japan) and added sequentially to water. To ensure complete dissolution, the mixture was stirred at 550 rpm using a magnetic stirrer for 1 h at room temperature (25°C). The solution was sterilized using a Nalgene™ Rapid-Flow™ Sterile Disposable Filter Unit (0.2 µm pore size; 431433; Thermo Scientific, MA, USA). iDisper was stored at −20 °C until use. Before use, the solution was thawed and heated to 37 °C in a water bath. The physicochemical properties were verified using a pH meter (FP20; Mettler Toledo, Switzerland) and an osmometer (Osmostat OM-6040; ARKRAY, Tokyo, Japan). The pH was measured to ensure it fell within the range of 5.5–6.0 without adjustment, and osmolality was adjusted to 295±5 mOsm/kg. To validate sterility, the filtered solution was plated on Soybean-Casein Digest (SCD) agar and incubated at 35°C. Colony-forming units (CFU) were quantified using an automatic colony counter (Scan500; Interscience, Saint Nom la Bretêche, France). Internal quality control records confirmed 0 CFU across independent manufacturing lots. iDisper was stored at −20 °C until use.

Raw Data Accessibility: Regarding the availability of data, we have uploaded the complete raw dataset underlying our statistical analyses as Supporting Information in this resubmission, rather than relying on the external ChemRxiv repository.

Comment 4: The interpretation of results appears exaggerated: Assertions of "comparable" performance to TrypLE Express (Figs 2-3) overlook significant differences in incubation duration, cell types, and metrics for passaging success. Figure 2d pie charts illustrate significant failure rates (approximately 30-50%) that cannot be accounted for by contamination or reagent-related problems. Examine the constraints of testing exclusively on birds and analyze the duration of papain allergens in a quantitative manner.

Response 4: Thank you for your comment. We have revised the manuscript to explicitly attribute the failure rates to raw material variability and provide a quantitative analysis of allergen stability based on data and literature.

Clarification of Figure 2d (Failure Rates): We revised the manuscript to explicitly attribute the failure rates to the current limitations of iDisper during the primary isolation phase. Clarification of Figure 2d (Failure Rates): We clarified that the failure rates shown in Figure 2d, particularly the 4.1% failure rate at passage 0 (P0), highlighted a specific challenge. Although iDisper is highly effective for cell expansion (

---

## [Decision Letter · Decision Letter 1]

24 Feb 2026

Dear Dr. Kawashima,

Thank you for submitting your manuscript to PLOS ONE. After careful consideration, we feel that it has merit but does not fully meet PLOS ONE’s publication criteria as it currently stands. Therefore, we invite you to submit a revised version of the manuscript that addresses the points raised during the review process.

A letter that responds to each point raised by the academic editor and reviewer(s). You should upload this letter as a separate file labeled ‘Response to Reviewers’.A marked-up copy of your manuscript that highlights changes made to the original version. You should upload this as a separate file labeled ‘Revised Manuscript with Track Changes’.An unmarked version of your revised paper without tracked changes. You should upload this as a separate file labeled ‘Manuscript’.

We look forward to receiving your revised manuscript.

Kind regards,

Shengqian Sun

Academic Editor

PLOS One

**Journal Requirements:**

Reviewers’ comments:

Reviewer’s Responses to Questions

**Comments to the Author**

Reviewer #1: (No Response)

Reviewer #2: All comments have been addressed

2. Is the manuscript technically sound, and do the data support the conclusions?

Reviewer #1: (No Response)

Reviewer #2: Yes

3. Has the statistical analysis been performed appropriately and rigorously?

Reviewer #1: (No Response)

Reviewer #2: Yes

4. Have the authors made all data underlying the findings in their manuscript fully available?

Reviewer #1: (No Response)

Reviewer #2: Yes

5. Is the manuscript presented in an intelligible fashion and written in standard English?

Reviewer #1: (No Response)

Reviewer #2: Yes

Reviewer #1: Abstract and Title: Refine the title for conciseness (e.g., "Food-Grade Cell Dissociation Agent via Regulatory Pre-Check Framework"). Expand abstract to explicitly state key metrics (e.g., 2 mg/mL papain efficacy, 8-month stability).

Introduction: Add 1-2 sentences citing recent cell-based food safety reviews (e.g., refs 3-4) to strengthen context on upstream challenges.

Methods: Specify exact iDisper osmolality adjustment protocol beyond "adjusted to 295±5 mOsm/kg" and include centrifuge RPM for all steps (e.g., 300 g noted inconsistently). Clarify embryo incubation humidity control details.

Results: In Fig 2d, quantify "4.1% failure rate" derivation (e.g., n= total cultures). Move S1 Appendix gene expression (S1 Fig) to main text as it supports phenotypic stability claims.

Discussion: Briefly address scalability for powder formulation (mentioned but not quantified) and potential mammalian cell testing limitations. Compare allergen management to FDA consultations (refs 11-15).

Figures/Tables: Ensure all scale bars are uniform (e.g., Fig 2a: 0.5 mm). Table 1 legend could note "Materials selected" more prominently.

Reviewer #2: (No Response)

.

Reviewer #1: No

Reviewer #2: No

---

## [Author Response · Author response to Decision Letter 2]

1 Mar 2026

We would like to express our sincere appreciation to the reviewers for their valuable comments and suggestions. We have carefully addressed all the points raised by the reviewers and have reflected them in the revised manuscript and the accompanying response letter.

---

## [Decision Letter · Decision Letter 2]

13 Mar 2026

A food-grade cell dissociation agent via regulatory pre-check framework

PONE-D-25-59887R2

Dear Dr. Kawashima,

We’re pleased to inform you that your manuscript has been judged scientifically suitable for publication and will be formally accepted for publication once it meets all outstanding technical requirements.

An invoice will be generated when your article is formally accepted. Please note, if your institution has a publishing partnership with PLOS and your article meets the relevant criteria, all or part of your publication costs will be covered. Please make sure your user information is up-to-date by logging into Editorial Manager at Editorial Manager® and clicking the ‘Update My Information’ link at the top of the page. For questions related to billing, please contact  and clicking the ‘Update My Information’ link at the top of the page. For questions related to billing, please contact  and clicking the ‘Update My Information’ link at the top of the page. For questions related to billing, please contact  and clicking the ‘Update My Information’ link at the top of the page. For questions related to billing, please contact billing support....

Kind regards,

Shengqian Sun

Academic Editor

PLOS One

Additional Editor Comments (optional):

Reviewers’ comments:

Reviewer’s Responses to Questions

**Comments to the Author**

Reviewer #1: (No Response)

2. Is the manuscript technically sound, and do the data support the conclusions?

Reviewer #1: (No Response)

3. Has the statistical analysis been performed appropriately and rigorously?

Reviewer #1: (No Response)

4. Have the authors made all data underlying the findings in their manuscript fully available?

Reviewer #1: (No Response)

5. Is the manuscript presented in an intelligible fashion and written in standard English?

Reviewer #1: (No Response)

Reviewer #1: I recommend that the revised version be accepted for publication in its current form, as it satisfactorily meets the academic and technical requirements expected for inclusion in the journal.

.

Reviewer #1: No

---

## [Editor Report · Acceptance letter]

PONE-D-25-59887R2

PLOS One

Dear Dr. Kawashima,

I’m pleased to inform you that your manuscript has been deemed suitable for publication in PLOS One. Congratulations! Your manuscript is now being handed over to our production team.

Lastly, if your institution or institutions have a press office, please let them know about your upcoming paper now to help maximize its impact. If they’ll be preparing press materials, please inform our press team within the next 48 hours. Your manuscript will remain under strict press embargo until 2 pm Eastern Time on the date of publication. For more information, please contact onepress@plos.org.

Kind regards,

on behalf of

Dr. Shengqian Sun

Academic Editor

PLOS One